# Functional Role of Hepatitis C Virus NS5A in the Regulation of Autophagy

**DOI:** 10.3390/pathogens13110980

**Published:** 2024-11-08

**Authors:** Po-Yuan Ke, Chau-Ting Yeh

**Affiliations:** 1Department of Biochemistry and Molecular Biology, Graduate Institute of Biomedical Sciences, College of Medicine, Chang Gung University, Taoyuan 33302, Taiwan; 2Liver Research Center, Chang Gung Memorial Hospital, Taoyuan 33305, Taiwan; chauting@cgmh.org.tw

**Keywords:** HCV, autophagy, selective autophagy, chaperone-mediated autophagy, microautophagy

## Abstract

Many types of RNA viruses, including the hepatitis C virus (HCV), activate autophagy in infected cells to promote viral growth and counteract the host defense response. Autophagy acts as a catabolic pathway in which unnecessary materials are removed via the lysosome, thus maintaining cellular homeostasis. The HCV non-structural 5A (NS5A) protein is a phosphoprotein required for viral RNA replication, virion assembly, and the determination of interferon (IFN) sensitivity. Recently, increasing evidence has shown that HCV NS5A can induce autophagy to promote mitochondrial turnover and the degradation of hepatocyte nuclear factor 1 alpha (HNF-1α) and diacylglycerol acyltransferase 1 (DGAT1). In this review, we summarize recent progress in understanding the detailed mechanism by which HCV NS5A triggers autophagy, and outline the physiological significance of the balance between host–virus interactions.

## 1. Introduction

Infection with RNA viruses, such as HCV, activates host cellular autophagy to promote viral growth and facilitate escape from antiviral defenses [1,2,3,4,5,6,7]. However, how HCV infections regulate autophagy remains controversial. Several HCV viral proteins, such as nonstructural (NS) 5A, may induce autophagy [8,9,10], but their functional roles in regulating autophagy are still unclear. Autophagy is an evolutionarily degradative process that eliminates unnecessary cytosolic materials via lysosomal acidic hydrolases [11,12,13,14,15,16,17]. The precise and proper control of autophagy is critical to maintain cellular homeostasis and ensure the recycling of nutrients [11,12,13,17,18,19,20,21,22]. Chronic HCV infection is the leading cause of late-stage liver disease [19,23,24,25,26,27]. Although current direct-acting antiviral drugs (DAAs) used for treating HCV infection are curative in more than 90% of patients [19,28,29,30,31], the detailed molecular mechanism by which HCV infection induces the development of chronic liver diseases remains unclear. Autophagy is categorized into three types: macroautophagy [11,12,17,32], chaperone-mediated autophagy (CMA) [33,34,35], and microautophagy [36,37,38,39,40]. All of these types of autophagy are critical for regulating the turnover of intracellular components, and their deregulation may develop human diseases [16,20,32]. Interestingly, HCV NS5A protein was recently shown to induce host mitophagy, CMA, and microautophagy, thus eliminating mitochondria, HNF-1α, and DGAT1 [8,9,10], implying that HCV NS5A could be a key player in the regulation of mitochondrial physiology and cellular metabolism in infected cells. In this review, we outline the current knowledge of how HCV NS5A regulates autophagy to promote mitochondrial turnover and protein degradation. Additionally, the regulatory mechanisms involved in these processes are summarized. Finally, the physiological significance and future directions of HCV NS5A-induced autophagy are discussed.

## 2. HCV

### 2.1. HCV Epidemiology

Chronic HCV infection in patients is a risk factor for late-stage liver diseases, which ultimately develop into hepatocellular carcinoma (HCC) [19,23,24,25,26,27]. Over 3% of the human population has been infected by HCV, and most of infected individuals progress to chronic liver diseases, generating a global public health burden [19,23,24,25,26,27]. A standard treatment regimen consisting of pegylated interferon-α and ribavirin has been used to control HCV infection for more than twenty years; however, the side effects and low efficacy of this regimen lead to treatment failure in infected individuals [41,42,43]. In the early 2010s, DAAs against HCV genome-encoded protease, phosphoprotein, and RNA-dependent RNA polymerase were developed and approved for the clinical treatment of HCV; these DAAs are curative in approximately 90% of patients [19,28,29,30,31,43]. However, the successful eradication of HCV via treatment with DAAs in infected individuals is hampered by high costs and the emergence of drug-resistant variants after treatment [14,21,28,30,44,45,46].

### 2.2. Organization of the HCV Genome

HCV is an RNA virus containing an envelope and belongs to the *Hepacivirus* genus of *Flaviviridae* family [19,23,25]. The HCV genome comprises a single-stranded, positive-sense RNA with a length of approximately 9.6 kilobases (Kb) [25,47,48]. The RNA genome of HCV is composed of a single open reading frame (ORF) that is flanked by 5′ and 3′ untranslated regions (UTRs) [47,48] (Figure 1). The HCV 5′-UTR contains an internal ribosome entry site (IRES) and a microRNA 122 (miR122)-binding site for activating translation and genome replication [25,47,48,49]. The 3′-UTR of HCV RNA increases IRES-mediated translation and harbors pathogen-associated molecular patterns (PAMPs) that trigger an innate antiviral response [50,51]. The HCV ORF encodes a single polyprotein of approximately 3000 amino acids (a.a.) [47,48] (Figure 1). The translated polypeptide is cleaved by cellular and viral proteases, generating three structural proteins (core, envelope glycoprotein 1 [E1], E2, and P7 proteins) and seven NS proteins (p7, NS2, NS3, NS4A, NS4B, NS5A, and NS5B proteins) [47] (Figure 1). The structural proteins are responsible for the assembly of infectious HCV particles, whereas the NS proteins participate in HCV replication and assembly of the HCV virion [47] (Figure 1).

### 2.3. Life Cycle of HCV

HCV primarily infects human hepatocytes and relies on several entry cofactors on the plasma membrane [52,53,54,55,56,57,58,59]. Circulating HCV in the bloodstream associates with lipoproteins, forming lipo-viral particles (LVPs) [60], which initially bind attachment factors on the cell surface, including low-density lipoprotein receptors (LDLRs) [56], scavenger receptor class B member I (SR-BI) [61], and heparan sulfate proteoglycans (HSPGs) [62]. Later, the HCV E2 protein directly interacts with SR-BI and the tetraspanin CD81 [61,63], leading to epidermal growth factor receptor (EGFR)-mediated activation of HRas [57,64]. HRas-induced actin rearrangement then promotes the interaction of CD81 with claudin 1 (CLDN1) [53,65], facilitating HCV internalization via clathrin-mediated endocytosis [66,67]. In addition, Occludin (OCLN), transferrin receptor 1 (TfR1), and the cholesterol receptor Niemann—Pick C1-like L1 (NPC1L1) function in the uptake and entry of HCV [54,58,59]. HCV internalization within acidic endocytotic vesicles drives low pH-dependent fusion of the HCV envelope with the endosome membrane, allowing the viral RNA genome to be released into the cytoplasm [68,69]. This viral RNA can be used as a template for the translation of viral proteins and replication of viral RNA. Finally, the newly synthesized viral genome and structural proteins assemble into infectious HCV virions, which then egress from cells [68,69,70].

## 3. HCV NS5A

### 3.1. The Domain Structure of NS5A

The HCV NS5A protein comprises an amphipathic α-helix (AH) at its N-terminus that associates with intracellular membranes, such as the endoplasmic reticulum, and three domains (domains I, II, and III) that are separated by two low-complexity sequences (LCSs, LCSI and LCSII) [71] (Figure 2). HCV NS5A domain I contains four conserved cysteine (Cys) residues capable of binding a single zinc atom, suggesting that HCV NS5A is a metalloprotein that coordinates zinc, which may regulate HCV RNA replication [72,73] (Figure 2). Unlike NS5A domain I, which is highly structured, domains II and III of HCV NS5A are not intrinsically ordered (Figure 2). NS5A domain II is required to replicate the HCV genome [74,75], whereas NS5A domain III participates in the assembly of the HCV virion, rather than in viral RNA replication [76,77] (Figure 2). The interferon sensitivity-determining region (ISDR) and protein kinase R-binding domain (PKRBD), which are located within the C-terminus of LCSI and the N-terminal region of domain II of NS5A, can regulate the sensitivity of the antiviral IFN response [78,79,80,81] (Figure 2). Putative serine (Ser) residues within LCSI may regulate HCV RNA replication [82,83,84]. The polyproline motifs within LCSII can interact with cellular SH3 domains and regulate viral RNA replication and HCV virion assembly [85].

### 3.2. The Hyperphosphorylation of NS5A

HCV NS5A is a hyperphosphorylated protein containing several Ser and threonine (Thr) residues that can be phosphorylated by intracellular kinases (Figure 2). Phosphorylation of Ser146 in the NS5A domain I negatively regulates the hyperphosphorylated status (p58) of NS5A by reducing the phosphorylation of Ser222 within the LCSI [86]. Phosphorylation at Ser229, Ser232, Ser235, and Ser238 in LCSI is necessary to replicate the HCV genome [82,83,84,87,88]. The phosphorylation of HCV NS5A at Ser225 may facilitate the interaction of NS5A with nucleosome assembly protein 1-like protein 1 (NAP1L1), bridging integrator 1 (Bin1), and vesicle-associated membrane protein-associated protein A (VAPA), thus regulating HCV NS5A localization and viral genome replication [89]. Notably, a recent report revealed that through its ATP-binding domain, HCV NS3A drives the sequential phosphorylation of HCV NS5A at Ser225, Ser232, and Ser235 [90]. The phosphorylation of Thr356, but not Thr348, within LCSII plays a role in HCV RNA replication [86,91]. In addition, ablation of phosphorylation at Ser408, Ser412, Ser414, Ser415, Ser452, Ser454, and 457 within NS5A domain III abolishes infectious particle production [92,93], suggesting that in addition to HCV RNA replication, NS5A phosphorylation regulates HCV assembly. In addition to the Ser and Thr residues, a recent study revealed that tyrosine (Tyr) residue 330 can be phosphorylated, which regulates HCV virion assembly [94]. Several kinases, including cAMP-dependent protein kinase A (PKA), casein kinases I and II (CK-I and CK-II), polo-like kinase (PLK), protein tyrosine kinase, and Abelson murine leukemia viral oncogene homolog 1 (Abl), are responsible for HCV NS5A phosphorylation [84,90,91,94,95,96].

### 3.3. Functional Role of NS5A in the HCV Life Cycle

Although no specific enzymatic activity of NS5A has been identified, NS5A is considered a multifunctional protein that may regulate HCV RNA replication by binding viral RNA, NS4B, and NS5B [97,98,99,100,101]. Additionally, HCV NS5A can interact with several host cellular factors, including prolyl-peptidyl isomerase, cyclophilin A (CypA) [102,103], VAPA [104,105], phosphatidylinositol-4-kinase IIIα (PI4KIIIα) [106,107], and rab GTP-binding protein 18 (Rab18) [108], to regulate the replication of viral RNA by increasing the binding affinity of NS5A for viral RNA and promoting the formation of a replication compartment, i.e., the “membranous web”. In addition to regulating replication, HCV NS5A may interact with the core to regulate virion production [92]. Moreover, the binding of HCV NS5A to DGAT1 and cortactin facilitates the interaction between HCV NS5A and the core of lipid droplets (LDs) to promote the assembly of infectious particles [109,110]. Since the essential role of NS5A lies in viral RNA replication and assembly of the HCV virion, the suppression of NS5A represents a target for developing DAAs for treating HCV [111,112,113]. Several DAAs against HCV NS5A, such as daclatasvir and ledipasvir, have been used in the clinic to cure HCV [111,112,113]. The persistence of several NS5A resistance-associated substitutions (RASs) in patients after DAA treatment failure significantly reduces the efficacy of DAA treatment [42,46].

## 4. Autophagy

### 4.1. Classification of Autophagy

Three forms of autophagy are classified as follows: macroautophagy, microautophagy, and CMA (Figure 3). Macroautophagy (also generally referred to as “autophagy”) involves stepwise vacuole biogenesis to promote the sequestration of intracellular materials within autophagosomes and the delivery of these materials to lysosomes for degradation [11,12,17,32] (Figure 3). Microautophagy involves the invagination and protrusion of the endosome and lysosome membrane to sequester a portion of the cytosol; this process produces intraluminal vesicles in which engulfed materials are degraded [36,37,38,39,40] (Figure 3). Microautophagy can be classified into two forms: fission-type microautophagy, which requires endosomal sorting complexes required for transport (ESCRT)-mediated membrane scission, and fusion-type microautophagy, which involves invagination and extension of the membrane (Figure 3). CMA is a selective process that consists of the recognition of cytosolic cargoes harboring the “Lys-Phe-Glu-Arg-Gln” (KFERQ) pentapeptide motif by the heat shock protein 70 kDa (HSC70, a cytosolic chaperone protein) and the delivery of these cargoes to lysosomes by a lysosomal membrane protein 2A (LAMP2A)-mediated docking process, ultimately leading to the elimination of engulfed cargoes [33,34,35] (Figure 3). Precise control of these three types of autophagy is essential for maintaining cell homeostasis, and their dysregulation may contribute to the development of human diseases [16,17,20,32].

### 4.2. Induction of Autophagy

The successful completion of autophagy requires the precise regulation of vacuole biogenesis, and action requires the regular functions of autophagy-related genes (ATGs) and the arrangement of intracellular membranes. The entire process of autophagy includes (1) the generation of a “cup-shaped″ isolation membrane (IM)/phagophore, (2) closure of the IM/phagophores to generate double-membrane autophagosomes, (3) fusion of autophagosomes with lysosomes to generate mature autolysosomes, and (4) lysosome reformation and recycling of autophagosomal components at termination [11,12,13,17,32,114,115,116,117] (Figure 3). Under stress, such as nutrient deficits in mammalian cells, repressed mammalian target of rapamycin complex 1 (mTORC1) activates the unc-51 like-kinase (ULK) complex (ULK1/2, FIP200/RB1CC1, ATG101, and ATG13), inducing the translocation of the ULK complex from the cytosol to the ER-associated membrane [118,119,120,121]. The resident ULK complex at the ER subdomain triggers the activation and recruitment of class III phosphatidylinositol-3-OH kinase (PI3KC3) complex I (PI3KC3/Vps34, PI3KR4/Vps15, ATG14, and Beclin 1), thereby promoting the production of phosphatidylinositol-3-phosphate (PtdIns(3)P) [115,122,123,124]. Newly generated PtdIns(3)P induces the recruitment of double-FYVE-containing protein 1 (DFCP1) and WD-repeat domain PtdIns(3)P-interacting (WIPI) family proteins, promoting the emergence of IM/phagophores (known as cradle-like structures) from the ER via lipid transfer and membrane tethering [115,122,123,125,126]. Other effectors involved in lipid transfer, membrane reconstitution, and IM/phagophore dissociation from the ER also play roles in IM/phagophore formation [127,128,129,130,131,132,133,134]. In addition to the ER, membrane resources from other organelles, including mitochondria, endosomes, the Golgi apparatus, the plasma membrane, and the mitochondria-associated ER membrane (MAM), may also support IM/phagophore formation [135,136,137,138,139,140,141,142].

### 4.3. Maturation of Autophagosomes

Elongation of the IM/phagophores and their closure to form mature autophagosomes require ubiquitin-like (UBL) conjugation events, including ATG12-ATG5-ATG16 complex conjugation and the attachment of phosphatidylethanolamine (PE) to the ATG8/microtubule-associated protein light chain 3 (LC3) family of proteins covalently (referred to as “ATG8/LC3-PE conjugation” and “ATG8/LC3 lipidation”) [121,143,144,145,146,147]. The enzyme activity of the ubiquitin-like activating enzyme 1 (E1) protein, ATG7 and the ubiquitin-like conjugation enzyme 2 (E2) protein, ATG10 promotes the covalent binding of ATG12 to ATG5, forming an ATG12-ATG5 conjugate. Then, ATG16 binds to ATG12-ATG5, ultimately generating the ATG12-ATG5-ATG16 complex [143,144,148,149,150]. ATG8/LC3 family proteins are posttranslationally processed by ATG4 family protein-mediated cleavage of their C-termini, producing the ATG8/LC3-I form. The conjugation of PE to ATG8/LC3-I subsequently generates ATG8/LC3-II (also known as ATG8/LC3-PE) through the ubiquitin ligase (E3) activity of the ATG12-ATG5-ATG16 complex [151]. Finally, ATG8/LC3-II facilitates membrane fusion to drive IM/phagophore expansion and, ultimately, its enclosure to form double-membrane autophagosomes [146,152,153], in which unwanted intracellular components are sequestered.

### 4.4. Fusion of Autophagosomes with Lysosomes

Autophagosome—lysosome fusion promotes the formation of autolysosomes, which enable lysosomal acidic hydrolases to degrade engulfed materials, thus promoting the recycling of nutrients [147,154,155,156,157,158]. Several functional molecules involved in vesicle trafficking, membrane tethering, and membrane fusion participate in the autophagosome—lysosome fusion [147,154,155,156,157,158]. Two soluble N-ethylmaleimide-sensitive factor attachment protein receptor (SNARE) complexes, the syntaxin 17 (STX17)-synaptosome-associated protein 29 (SNAP29)-vesicle-associated membrane protein 7 (VAMP7)/VAMP8 complex [159,160] and the YKT6-SNAP29-STX7 complex [161,162], facilitate membrane fusion. Tethering factors, including the homotypic fusion and protein sorting (HOPS) complex [15], ATG14 [18], and tectonin β-propeller repeat containing 1 (TECPR1) [163] promote the assembly of SNARE complexes during autophagosome—lysosome fusion. Other tethers that similarly function in the fusion of autophagosomes with lysosomes, such as pleckstrin homology domain-containing family M member 1 (PLEKHM1) were also reported recently [15,22,164,165,166,167]. The Rab family of small GTPases [168,169,170,171,172], cytoskeletal motor proteins [171,172,173,174,175,176,177], and cytoskeleton polymerization–mediated minus end-directed and retrograde movement on microtubules [171,178] enable the perinuclear transport of autophagosomes for their fusion with lysosomes.

### 4.5. Termination of Autophagy

After the process of autophagy is complete, nutrient restoration reactivates mTOR, leading to autophagy termination and autophagic lysosome reformation (ALR) [179,180]. The clathrin-mediated endocytosis regulator (clathrin and adaptor protein 2 [AP2]), the lysosomal efflux permease, spinster, the ubiquitin E3 ligase Cullin 3-Kelch-like protein 20 (KLHL20), and kinesin heavy chain family protein 5B (KIF5B) function in the regulation of ALR. In addition to ALR, autophagosomal component recycling (ACR) was recently shown to function in the termination of autophagy [181,182]. The recycler complex, which is composed of sorting nexins (SNXs, including SNX4, SNX5, and SNX17), promotes the recycling of STX17 and ATG9 via ACR [181,182].

## 5. Selective Autophagy

### 5.1. The Process of Selective Autophagy

Autophagy not only serves as a bulk, nonselective degradation process; increasing studies indicate that autophagy may specifically target cargoes, including intracellular organelles and proteins, for degradation, so-called “selective autophagy” [183,184,185,186,187]. The sequestration of degradative cargoes for selective autophagy relies on the recognition of polyubiquitinated proteins by cargo receptors and interactions between cargoes and adaptor proteins [186,187,188,189,190,191,192,193,194] (Figure 3). Many cargo receptors that play a role in selective autophagy, such as p62/sequestosome 1 (p62/SQSTM1), calcium-binding and coiled-coil domain-containing protein 2 (Calcoco2/NDP52), and optineurin (OPTN) have been identified [186,187,190,191,193,194]. The ubiquitin-associated (UBA) domain and LC3-interacting regions (LIRs) within most of these cargo receptors bridge their interaction with ubiquitinated cargoes and binding to ATG8/LC3 family proteins, respectively [186,187,190,191,193,194]. Moreover, two additional GABARAP-interacting motifs and ATG8-interacting motifs within cargo receptors also facilitate the recognition of degradative cargoes in selective autophagy [195,196,197,198].

### 5.2. Organellophagy

Selective autophagy can eliminate intracellular organelles; this process, referred to as “organellophagy”, involves the removal of damaged organelles and thus facilitates organelle regeneration. Organellophagy can specifically target mitochondria, ER, peroxisomes, lysosomes, ribosomes, nuclei, and LDs for degradation, thereby maintaining organelle integrity [183,185,199]. The regular control of organellophagy promotes organelle regeneration and maintains intracellular metabolic balance.

The removal of deformed mitochondria by organellophagy, called “mitophagy”, represents quality control to regulate mitochondrial integrity [200,201,202]. Mitochondrial depolarization and fission inhibit presenilin-associated rhomboid-like protein (PARL)-mediated cleavage of PTEN-induced putative kinase 1 (PINK1), stabilizing PINK1 on the mitochondrial outer membrane (MOM) [203,204]. The accumulation of PINK1 on the MOM, in turn, induces the recruitment of the ubiquitin E3 ligase Parkin and the phosphorylation of ubiquitin and Parkin at Ser65 [205,206,207,208,209,210]. The translocation of Parkin to mitochondria subsequently triggers the polyubiquitination of MOM proteins [205,206,207,208,211]; this process thus recruits mitophagy receptors, including OPTN and NDP52/Calcoco2, for PINK1/Parkin-dependent mitochondrial turnover [201,212]. In addition, other cargo receptors that analogously drive mitophagy have also been identified recently [213,214,215,216,217,218,219,220,221,222,223,224,225,226]. Moreover, phosphorylation of mitophagy receptors by tank-binding kinase 1 (TBK1) may promote PINK1/Parkin-dependent mitophagy [227,228,229]. Additionally, the recruitment of DFCP1 and WIPI family proteins, the generation of PtdIns(4,5)P_2,_ and F-actin polymerization at the region proximal to degradative mitochondria facilitate IM/phagophore formation for efficient mitochondrial removal via mitophagy [212,230].

## 6. HCV and Autophagy

### 6.1. Induction of Autophagy by HCV

HCV was first found to activate autophagy in immortalized HHs (IHHs) harboring the HCV H77 (belonging to genotype 1a) genome [231] (Table 1). Later, the transfection of HCV JFH1 belonging to genotype 2a) RNA in Huh7 human hepatoma cells was shown to induce incomplete autophagy through the unfolded protein response (UPR), and UPR-induced autophagy is required for the replication of HCV [232] (Table 1). The infection of Huh7.5.1 cells with HCV JFH1, a Huh7-derived clone lacking a competent retinoic acid-inducible gene-I (RIG-I) antiviral response, triggers autophagy, which regulates the initial translation of HCV RNA [233,234,235,236] (Table 1). Moreover, the infection of Huh7 cells with HCV JFH1 was demonstrated to activate complete autophagy to suppress HCV PAMP-induced innate antiviral immunity, thus promoting HCV replication [236,237] (Table 1). Similarly, infection of IHHs with HCV H77 also induces autophagy to repress the IFN antiviral response [236,237,238] (Table 1). The inhibition of antiviral immunity by HCV-induced autophagy was further supported by the finding that tumor necrosis factor receptor (TNFR)-associated factor 6 (TRAF6) degradation is induced through the autophagy receptor p62/SQSTM1 in infected cells [239] (Table 1). In addition, HCV-induced autophagy reportedly impairs IFN-α-triggered innate immune signaling by downregulating the level of IFN-α receptor 1(IFNAR1) [240] (Table 1). These findings are also supported by Mori et al.’s study, which revealed that the replication of HCV JFH1 induces selective autophagy in Huh7.5.1 cells [241] (Table 1). In addition, HCV JFH1 infection activates autophagy to promote the turnover of mitochondria, LDs, and viral proteins [9,242,243,244] (Table 1). Mitochondrial degradation by Parkin-dependent mitophagy in HCV-infected cells prevents cell death and promotes persistent virus infection [243,245] (Table 1). Given that the quality control of mitochondria by mitophagy plays a crucial role in controlling innate immunity [246,247,248,249,250], it remains unclear whether HCV-induced mitophagy regulates HCV PAMP-triggered innate antiviral response. Additionally, HCV-induced autophagy could participate in the catabolism of LDs through an unresolved mechanism [242] (Table 1). In addition to innate antiviral immunity, HCV infection has also been shown to activate the nucleotide-binding domain, leucine-rich-containing family, pyrin domain–containing-3 (NLRP3) inflammasome, which leads to the release of interleukin 1 beta (IL-1β) [251,252,253]. Although autophagy and mitophagy have been shown to regulate the activation of the NLRP3 inflammasome [254,255], whether HCV-induced autophagy and mitophagy modulate the NLRP3 inflammasome response remains uncertain. These studies imply that HCV-activated autophagy and mitophagy may interfere with innate antiviral immunity, alter the homeostasis of lipid metabolism, and regulate the inflammatory response.

Several studies have shown that HCV triggers the biogenesis of autophagosomes [256,257,258,259,260]. The induction of autophagosomes by HCV induces the regeneration of membranous compartments to replicate HCV viral RNA [259] (Table 1), mainly through the interaction of ATG5 with NS4B and NS5B transiently [260] (Table 1). Molecules that function in the initial stage of autophagy, including PI3KC3/Vps34 and DFCP1, reportedly positively regulate the replication of viral RNA via the formation of a membranous structure [256] (Table 1). At the initial infection stage, HCV can induce the expression of the run domain Beclin-1-interacting and cysteine-rich domain-containing protein (Rubicon) and repress UV irradiation resistance-associated gene protein (UVRAG) expression to delay the maturation of autophagosomes, which support HCV viral RNA replication [258] (Table 1). In addition, HCV infection induces the emergence of phagophores from the ER and triggers STX7-mediated homotypic fusion for the biogenesis of autophagosomes, thus supporting the replication of HCV viral RNA [257] (Table 1). On the other hand, autophagy regulates the HCV virion assembly in infected cells [261] (Table 1). HCV-induced autophagy may promote the egress of infectious virions through apolipoprotein E (ApoE) transport and the CD63-associated exosome pathway [262,263] (Table 1). These studies indicate that virus-induced autophagy plays a proviral role in the HCV life cycle.

**Table 1 pathogens-13-00980-t001:** Regulation of autophagy by HCV.

Approach	Phenotype	Functional Role	Reference(s)
Infection of IHHs with HCV H77 (1a)	Formation of GFP-LC3-labeled autophagic vacuoles Accumulation of autophagic vacuoles observed in TEM micrographs	Promote viral RNA replication	[231]
Transfection of HCV JFH1 (2a) RNA in Huh7.5 cells	Increased LC3B-II levels Increased formation of autophagosomes, which did not fuse with lysosomes Activation of incomplete autophagy by the UPR	Promote viral RNA replication	[232]
Infection of Huh7 cells with HCV JFH1	Increased LC3B-II levels Formation of GFP-LC3-labeled autophagic vacuoles No detectable colocalization between autophagic vacuoles and viral proteins	Promote the translation of incoming viral RNA	[236]
Infection of Huh7 cells with HCV-JFH1	Accumulation of autophagosomes and autolysosomes in infected cells Enhanced autophagic flux by HCV infection Activation of complete autophagy by the UPR	Promote viral RNA replication/repress innate antiviral immunity	[237]
Infection of IHHs with HCV-H77 and HCV-JFH1	Increased interferon response in HCV-infection IHHs by interference with autophagy Activation of caspase-dependent apoptosis by autophagy deficiency in HCV-infected IHHs	Promote viral RNA replication/repress innate antiviral immunity	[238]
Infection of Huh7 cells with HCV-JFH1	Induced TRAF6 degradation by HCV infection Interaction between TRAF6 and p62/SQSTM1 The essential role of p62/SQSTM1 in TRAF6 degradation	Promote viral RNA replication/repress innate antiviral immunity	[239]
Infection of Huh7 and Huh7.5 cells with HCV-JFH1	Impaired IFN-α-mediated antiviral response by persistent HCV infection Downregulation of IFNAR1 by persistent HCV replication Restored IFN-α-mediated anti-HCV immunity by inhibition of autophagy in persistently infected cells	Suppress IFN-α-triggered innate antiviral signaling	[240]
HCV-JFH1 infection and HCV JFH1 subgenomic RNA transfection in Huh7 and Huh7.5.1 cells	Increased the localization of ubiquitinated compartments with LC3-positive puncta Requirement of ATG5 and ATG14 for HCV-induced autophagy The replication of HCV RNA was sufficient to activate selective autophagy	Unclear	[241]
Transfection of HCV Con1 (1b) and HCV JFH1 subgenomic RNA in Huh7 and Huh7.5-1 cells	An inverse correlation between steatosis and autophagy Colocalization between LDs and autophagic vacuoles Accumulation of cholesterol via impaired autophagy	Promote LD catabolism	[242]
HCV-JFH1 infection and transfection of Con1 subgenomic RNA in Huh7.5.1 cells	Mitochondrial depolarization induced by HCV infection Mitochondrial translocation of Parkin induced by HCV infection Increased PINK1 expression and the ubiquitination of Parkin and mitochondrial proteins	Protect infected cells from apoptosis/establish viral persistence	[243,245]
Transfection of HCV JFH1 RNA in Huh7.5 cells	Colocalization of NS5A and NS5B within nascent viral RNA with autophagosomes The essential roles of LC3 and ATG7 for replicating HCV RNA Autophagosomal membranes acting as the site of HCV RNA replication	Promote viral RNA replication	[259]
Infection of Huh7 cells with HCV-JFH1	Increase in the interaction of ATG5 with NS5B and NS4B induced by HCV infection The essential role of ATG5 to establish replication compartments	Promote viral RNA replication	[260]
Transfection of HCV Con1 (1b) and HCV JFH1 subgenomic RNA in Huh7/Infection of Huh7 cells with HCV JC1 (2a)	Suppressed replication of HCV viral RNA by inhibition of PI3KC3/Vps34 using wortmannin Inhibited HCV viral RNA replication by gene silencing of PI3KC3/Vps34 and DFCP1 Transient association of DFCP1 with the nascent replication complexes of HCV viral RNA	Promote viral RNA replication	[256,259]
Infection of Huh7 and Huh7.5 cells with HCV-JFH1	Delayed maturation of autophagosome by HCV infection Promoted viral replication and repressed maturation of autophagosome by Rubicon in HCV-infected cells Repressed viral replication and enhanced maturation of autophagosome by UVRAG in HCV-infected cells Induced Rubicon expression by HCV NS4B	Promote viral RNA replication	[258]
Infection of Huh7 and Huh7.5 cells with HCV-JFH1	Induced homotypic fusion of phagorphores via STX7 by HCV infection Emergence of phagophores at the ER by HCV infection Phagophores as the replication sites of HCV viral RNA	Promote viral RNA replication	[257]
Infection of Huh7.5-1 cells with HCV-JFH1	Accumulation of GFP-LC3-labeled autophagic vacuoles No significant colocalization between autophagic vacuoles and viral proteins	Enhance the assembly of viral particles	[261]
Infection of IHHs and Huh7.5 cells with HCV-H77 and HCV-JFH1	Increased autophagosome-lysosome fusion induced by HCV infection Accumulation of intracellular virions induced by autophagy interference in infected cells Reduced extracellular virions in infected cells induced by autophagy interference Suppressed release of exosome-associated virions induced by autophagy interference	Enhance the assembly of viral particles	[262]
Infection of Huh7 and Huh7.5-1 cells with HCV-JFH1	Colocalization of GFP-LC3-labeled autophagic vacuoles with apolipoprotein A (ApoE) in replicons and infected cells Autophagy-induced degradation of ApoE in infected cells Decreased extracellular levels of infectious particles in infected cells induced by autophagy interference	Enhance the assembly of viral particles	[263]

### 6.2. Activation of Autophagy by HCV Viral Proteins

Among the HCV genome-encoded proteins, NS4B was the first viral protein shown to induce incomplete autophagy in Huh7 cells through Rab5 and PI3KC3, which may promote HCV replication [264] (Table 2). HCV NS3 can trigger autophagy via an immunity-associated GTPase family M (IRGM)-dependent pathway to increase viral infectivity [265] (Table 2). In addition, the HCV core protein also induces autophagy via the UPR and eukaryotic translation initiation factor 2 alpha kinase 3 (EIF2AK3)-activating transcription factor 6 (ATF6) axis-mediated upregulation of ATG12 and LC3B [266] (Table 2). In contrast, the HCV core protein may repress mitophagy by interfering with Parkin translocation to mitochondria and later mitochondrial ubiquitination [267] (Table 2). Moreover, HCV NS5A also induces autophagy to promote mitochondrial turnover, HNF-1α degradation (Table 2), and DGAT1 degradation [9,10,268] (Table 2). These studies indicate that HCV proteins play specific roles in regulating autophagy.

## 7. Regulation of Autophagy by HCV NS5A

### 7.1. Induction of Mitophagy by HCV NS5A Eliminates Mitochondria

Siu and colleagues first reported that the transfection of HCV JFH1 RNA in Huh7 cells induces ER–mitochondrion tethering, which is characterized by the formation of a MAM that wraps around the mitochondrion [269] (Table 3). Additionally, the authors showed that ectopic expression of HCV NS5A induces the formation of ER-tethered mitochondria, to which HCV NS5A is mainly localized, in human embryonic kidney 293 (HEK293) cells [269] (Table 3). In addition, the overexpression of HCV NS5A induces mitochondrial fragmentation, regardless of the activity of dynamin-related protein 1 (Drp1) [269] (Table 3). Moreover, PI4KIIIα inhibitor treatment and overexpression of a PI4KIIIα kinase-dead (KD) mutant suppressed mitochondrial fragmentation in HCV NS5A-overexpressing cells [269] (Table 3), suggesting that PI4KIIIα kinase activity is required for HCV NS5A-induced mitochondrial fragmentation. Similarly, interference with the interaction between HCV NS5A and PI4KIIIα by ectopic expression of a PI4KIIIα mutant lacking an NS5A-interacting motif and HCV NS5A lacking the PI4KIIIα-binding domain repressed mitochondrial fragmentation in cells [269] (Table 3), suggesting that the interaction between HCV NS5A and PI4KIIIα is critical for the induction of mitochondrial fragmentation. The overexpression of PI4KIIIα KD and a dominant-negative mutant of Drp1 (K38D) inhibited mitochondrial fragmentation in Huh7.5.1 cells infected with HCV JC1 (belonging to genotype 2a) [269] (Table 3), indicating that both PI4KIIIα and Drp1 are necessary for HCV-induced mitochondrial fragmentation. Furthermore, the overexpression of HCV NS5A in cells alleviated hydrogen peroxide-induced cell apoptosis [269] (Table 3). Together, these studies suggest that HCV NS5A may induce mitochondrial fragmentation through its interaction with PI4KIIIα and that this process could prevent cells from undergoing apoptosis.

Soon afterward, Jassey et al. reported that ectopic expression of HCV NS5A in Huh-7.5 cells induces the formation of green fluorescence protein (GFP)-tagged LC3 puncta and increases the expression of PE-conjugated LC3, resembling the effects of rapamycin and carbonyl cyanide m-chlorophenyl hydrazine (CCCP) [8] (Table 3), suggesting that HCV NS5A expression alone is sufficient to activate autophagy. Staining with the dye JC-1, which acts as an indicator of the mitochondrial membrane potential, coupled with flow cytometry and immunofluorescence analysis revealed that the overexpression of HCV NS5A in Huh-7.5 cells induces depolarization and mitochondrial fragmentation [8] (Table 3), indicating that HCV NS5A expression is sufficient to drive mitochondrial deformation. In addition, biochemical and immunofluorescence analyses revealed that HCV NS5A overexpression triggers the recruitment of Parkin to mitochondria, which is similar to the effects of CCCP treatment [8] (Table 3), suggesting that HCV NS5A promotes mitochondrial translocation. The induced production of mCherry^+^/EGFP^+^ autophagosomes and mCherry^+^/EGFP^−^ autolysosomes containing the mCherry-EGFP-LC3 reporter and the increased accumulation of LC3-II induced by bafilomycin A1 in HCV NS5A-overexpressing cells suggest that HCV NS5A activates complete autophagy [8] (Table 3). Moreover, treatment with the antioxidant N-acetyl cysteine (NAC) attenuated the expression of total LC3-II and mitochondrial LC3-II in Huh-7.5 cells harboring HCV NS5A [8] (Table 3), indicating that suppression of reactive oxygen species (ROS) inhibits HCV NS5A-induced autophagy and mitophagy. Furthermore, overexpression of the HCV core protein in HCV NS5A-overexpressing Huh-7.5 cells diminishes the mitochondrial translocation of Parkin [8], suggesting that the HCV core protein interferes with HCV NS5A-induced mitophagy. These studies imply that HCV NS5A may drive mitophagy and that the HCV core protein may suppress this process. The physiological significance of HCV NS5A-activated mitophagy in the pathogenesis of HCV-related liver diseases remains largely unclear. Mitophagy has been implicated in the development of liver diseases, including HCC [270,271,272,273]. In particular, mitophagy activation was recently shown to promote HCC development by inducing the production of ROS, the stemness of cancer stem cells (CSCs), and the reprogramming of energy metabolism [274,275,276]. Therefore, the functional role(s) of HCV NS5A-induced mitophagy in the occurrence of HCV-associated HCC and its clinical relevance should be investigated in the future. Since mitophagy has emerged its role(s) in regulating viral PAMP-induced antiviral signaling and NLRP3 inflammasome activation, whether HCV NS5A activates mitophagy to modulate innate antiviral immunity and the inflammatory response in HCV-infected cells is interesting to study in the future.

### 7.2. HCV NS5A-Induced CMA Degrades HNF-1α

Matsui and colleagues reported that HNF-1α is a transcription factor that may transactivate the mRNA expression of glucose transporter 2 (GLUT2) during HCV replication [277]. The overexpression of HCV NS5A during HCV J6/JFH1 (belonging to genotype 2a) infection leads to lysosome-dependent HNF-1α degradation [277]. The degradation of HNF-1α within lysosomes induced by HCV may repress GLUT2 mRNA expression, thus interfering with glucose metabolism [277]. Matsui et al. further reported that ectopically expressed HNF-1α coimmunoprecipitated with endogenous and overexpressed HSC70, the chaperone involved in CMA, in Huh-7.5 cells [9] (Table 3). The authors reported that HNF-1α contains a putative pentapeptide (“Gln-Arg-Glu-Val-Val”) that can interact with the CMA chaperone HSC70 and that mutation of this motif diminished the binding of HNF-1α to HSC70 [9] (Table 3), suggesting that HNF-1α is a substrate of CMA-mediated degradation. The overexpression of HCV NS5A promotes the binding of HNF-1α to HSC70, and HCV JFH1 infection also increases the binding of HNF-1α to HSC70 [9] (Table 3). In addition to HNF-1α, HCV NS5A can bind HSC70 through its domain I (a.a. 121~126) [9] (Table 3), implying that HCV NS5A connects the interaction of HNF-1α with HSC70. Moreover, HCV J6/JFH1 infection leads to HNF-1α degradation, which is inhibited by gene knockdown of LAMP2A and HSC70, and by ammonia chloride treatment [9] (Table 3), suggesting that HCV infection may activate CMA to degrade HNF-1α. Immunofluorescence analysis revealed that HCV J6/JFH1 infection promoted the colocalization of HCV NS5A and HNF-1α [9] (Table 3). Additionally, ectopic expression of HCV NS5A leads to the redistribution of HNF-1α to lysosomes containing HCV NS5A [9] (Table 3), suggesting that HCV NS5A acts as a bridge to localize HNF-1α within lysosomes. Treatment with the lysosomal degradation inhibitor pepstatin A suppresses the lysosomal degradation of HNF-1α, and a mutation in the pentapeptide motif of HNF-1α (Gln130Ala) reduces the lysosomal translocation of HNF-1α [9] (Table 3). These studies indicate that HCV NS5A facilitates the recruitment of HNF-1α to HSC70 in lysosomes, thus inducing HNF-1α degradation via CMA. Somatic mutations in HNF-1α have been found in the cancerous tissues of 1~2% of HCC patients [278], and are highly related to cell stemness and liver malignancies [279], indicating that the HCV NS5A-regulated CMA-mediated HNF-1α could participate in the pathogenesis of HCV-associated HCC. However, further studies are needed to explore the clinical relevance of HCV NS5A-activated CMA in the development of HCC.

### 7.3. The Triggering of Endosomal Microautophagy by HCV NS5A Induces DGAT1 Degradation

DGAT1 is a critical enzyme that catalyzes the final step in triglyceride biosynthesis, which is necessary for the biogenesis of LDs [280]. Additionally, DGAT1 is the host cellular factor required for the efficient assembly of the HCV virion, presumably through its ability to enhance the interaction between HCV NS5A and the core protein on the LD surface [109,281]. Yuliandari and colleagues reported that ectopically expressed HCV NS5A coimmunoprecipitates with DGAT1, which is overexpressed in Huh-7.5 cells [10] (Table 3). Additionally, immunofluorescence analysis of these overexpressed cells showed that HCV NS5A and DGAT1 colocalize [10] (Table 3). The author reported that HCV NS5A interacts with DGAT1 through domain I of HCV NX5A (a.a. 1-213) and that a.a. 157~399 of DGAT1 are critical for the binding of DGAT1 to HCV NS5A [10] (Table 3). In addition, HCV J6/JFH1 infection leads to DGAT1 degradation, which is inhibited by ammonia chloride [10] (Table 3), suggesting that DGAT1 is a substrate of lysosomal degradation. Similarly, HCV NS5A overexpression also induces the degradation of DGAT1 [10] (Table 3). Moreover, DGAT1 contains a pentapeptide, “Gln-Val-Glu-Lys-Arg” (a.a. 149~153), that can interact with HSC70, a chaperone necessary for CMA and endosomal microautophagy (eMI) [10] (Table 3). A mutation within this pentapeptide (Gln149Ala) of DGAT1 interrupts the binding of DGAT1 to HSC70 [10] (Table 3), suggesting that DGAT1 specifically binds HSC70. The authors found, by immunofluorescence analysis, that DGAT1 colocalizes with HCV NS5A on lysosomes and late endosomes. Genetic knockdown of VPS4B, which functions in eMI but not LAMP2A, restores DGAT1 expression in HCV J6/JFH1-infected cells [10] (Table 3). Collectively, these studies revealed that eMI is the main route by which HCV induces the lysosomal degradation of DGAT1. Since DGAT1 has been shown to regulate LD biogenesis and HCV virion assembly [109,280,281], investigating whether HCV NS5A-induced degradation of DGAT1 via eMI may interfere with the biogenesis of LDs from excess free fatty acids (FFAs) and establish persistent infection in infected cells is valuable. Also, the physiological significance and clinical relevance of HCV NS5A-induced eMI in the pathogenesis of HCV-related liver steatosis require further investigation.

## 8. Conclusions and Perspectives

Several recent studies have suggested that HCV NS5A plays an unexpected role in regulating HCV-induced autophagy [8,9,10]. Notably, HCV NS5A induces autophagy to selectively eliminate mitochondria via mitophagy and target HNF-1α and DGAT1 for CMA- and eMI-mediated degradation, respectively (Figure 4). Importantly, these studies suggest a new paradigm, in which HCV NS5A alone is sufficient to trigger autophagic degradation, which may allow cells to counteract oxidative stress [8], reduce glucose metabolism [9], and alter LD biogenesis [10] (Figure 4). However, the detailed mechanism by which HCV NS5A induces these autophagy-related degradation processes is largely unknown. In the context of mitophagy, how HCV NS5A induces mitochondrial depolarization and promotes the mitochondrial translocation of Parkin is unclear. Since the HCV core protein has been shown to interrupt mitophagy in cells harboring HCV NS5A [8,273] and to affect the LD localization of HCV NS proteins [282], whether HCV NS5A has a similar phenotype in infected cells in which the complete HCV life cycle is carried out is questionable. Additionally, it remains unclear whether HCV NS5A, which is overexpressed, maintains a competent hyperphosphorylation status similar to that of NS5A, which is encoded by the entire HCV genome in infected cells. Moreover, the specific domain of HCV NS5A and whether the hyperphosphorylation of HCV NS5A can alter the ability of NS5A to activate mitophagy are also unknown. Further studies are urgently needed to determine whether and how the HCV NS5A phosphorylation status regulates mitochondrial turnover in HCV-infected cells.

Intriguingly, HCV NS5A is sufficient to activate CMA, which is accompanied mainly by macroautophagy activation in cells. HCV NS5A may direct the interaction of HNF-1α with HSC70 on lysosomes, and this process could differ from the LD-mediated relocalization of HCV NS5A mediated by the capsid protein in HCV-infected cells [282]. However, whether and how HCV NS5A is redistributed into various subcellular compartments to activate mitophagy, CMA, and eMI remain largely unknown. Additionally, little is known about how and why HCV infection simultaneously triggers these diverse autophagy processes through NS5A in infected cells. In this context, further investigations are urgently needed to delineate the underlying mechanism and depict the spatiotemporal control of HCV NS5A-mediated activation of autophagic degradation in cells in which the entire HCV life cycle can be carried out. Domains I and II of HCV NS5A are crucial to the replication of viral RNA [72,73,74,75], and domain III functions in virion assembly [76,77]. Therefore, it is interesting to study whether HCV NS5A may activate autophagy to promote virus replication by reconstituting intracellular membranes in the future. Additionally, whether HCV-induced autophagy regulates the stability and intracellular trafficking of intracellular proteins to enhance the assembly of infectious particles is worth further investigation. Furthermore, additional studies are needed to determine whether HCV NS5A-activated mitophagy, CMA, and eMI regulate HCV growth and participate in the development of HCV-related liver diseases. Also, the physiological significance and clinical relevance of HCV NS5A-induced mitochondrial turnover, CMA-induced degradation of HNF-1α, and eMI-mediated DGAT1 degradation in the pathogenesis of HCV-associated liver diseases, including liver steatosis and HCC, urgently need to be studied. In addition, whether HCV NS5A-induced mitophagy is involved in the regulation of innate antiviral immunity and inflammasome activation, thereby deregulating the immunological response and affecting clinical outcomes in HCV-infected patients, also needs to be understood in the future. On the other hand, it remains uncertain whether treatment with HCV DAAs against NS5A in infected patients suppresses the degradation of mitochondria, HNF-1α, and DGAT1 in HCV-infected hepatocytes. Many abnormal mitochondria and the ER remain in the liver tissues of HCV-infected patients after one year of treatment with DAAs [283], which suggests that pharmacological modulation of mitophagy, CMA, and eMI could be used to improve the clinical consequences of HCV infection after treatment with DAAs.

## Figures and Tables

**Figure 1 pathogens-13-00980-f001:**
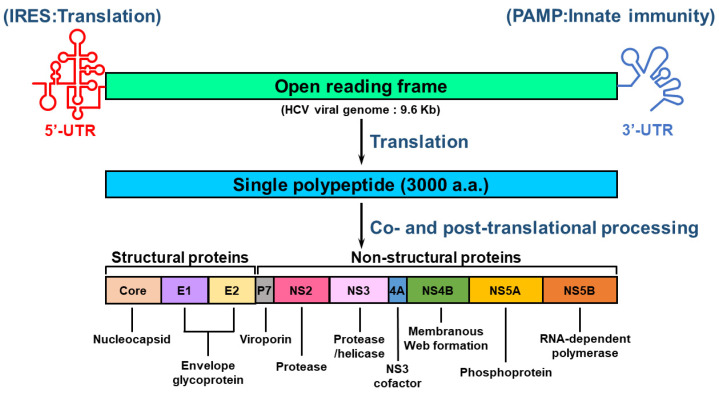
Genome organization of HCV. The viral genome of HCV consists of a positive-sense, single-stranded RNA with a length of approximately 9.6 kb. The viral genome of HCV includes an open reading frame (ORF) that is flanked by 5′- and 3′-untranslated regions (UTRs). The 5′-UTR contains an internal ribosome entry site (IRES) for stimulating translation, whereas the 3′-UTR harbors a pathogen-associated molecular pattern (PAMP) for inducing innate antiviral immunity. The ORF encodes a single polypeptide that is processed by proteases encoded by host cellular RNA and viral RNA to form structural proteins (core, E1, and E2) and nonstructural proteins (p7, NS2, NS3, NS4A, NS4B, NS5A, and NS5B). The core, E1, and E2 proteins are the major constituents of the HCV virion, and other viral proteins differentially regulate the replication of viral RNA and the assembly of infectious particles. The enzymatic activities of these viral proteins are indicated.

**Figure 2 pathogens-13-00980-f002:**
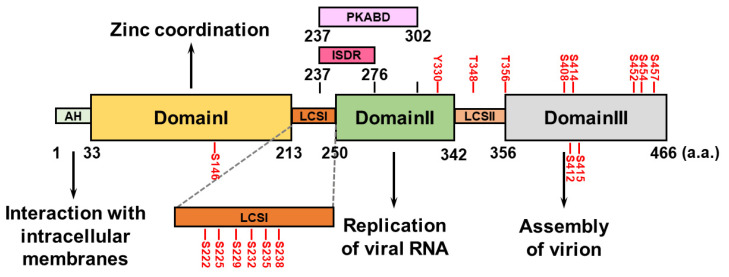
Domain structure of HCV NS5A. HCV NS5A contains an amphipathic helix (AH) at its N-terminal region and three specific domains: I, II, and III. They are flanked by two low-complexity sequences (LCSs), LCSI and LCSII. The functional role(s) of each domain in the HCV life cycle are specifically indicated. The interferon sensitivity-determining region (ISDR; a.a. 237~276) and the protein kinase R-binding domain (PKRBD; a.a. 237~302) are shown. The serine (S), threonine (T), and tyrosine (Y) residues within HCV NS5A that can be phosphorylated are indicated.

**Figure 3 pathogens-13-00980-f003:**
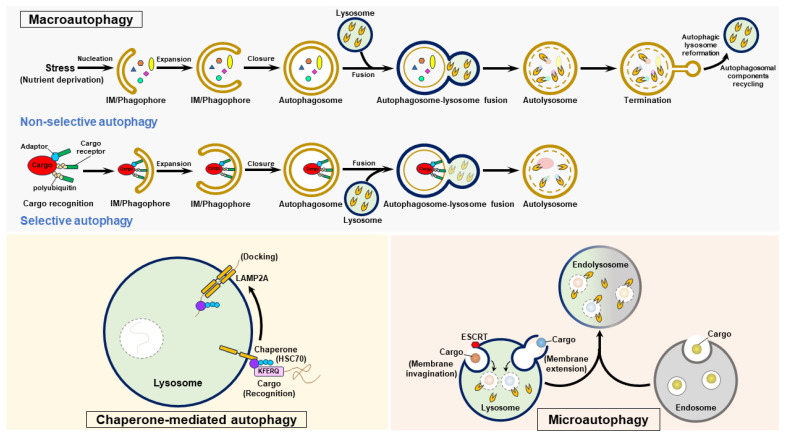
The autophagy process. Autophagy is categorized into three forms: macroautophagy, chaperone-mediated autophagy (CMA), and microautophagy. To initiate macroautophagy, various stressors, such as nutrient deprivation, induce nucleation of the isolation membrane (IM)/phagophore. The IM/phagophore expands and encloses cytosolic components to form mature autophagosomes surrounded by double membranes. The fusion of autophagosomes with lysosomes generates autolysosomes, and lysosomal hydrolases degrade the elements in the autolysosome interior. The termination of autophagy is coupled with the reformation of tubular lysosome (ALR) and autophagosomal component recycling (ACR), thus regenerating lysosomes and autophagic molecules. Selective autophagy is a specific form of macroautophagy that requires cargo receptors to recognize degradative cargoes via the ubiquitination of cargoes and other adaptor proteins, ultimately delivering these targeted proteins for autophagic degradation. CMA selectively targets degradative substrates harboring the pentapeptide “KFERQ”, which is recognized by HSC70. Through its binding to LAMP2 on the lysosomal membrane, the substrate-HSC70 complex is delivered into lysosomal lumen for degradation. Microautophagy occurs in endosomes and lysosomes, and involves the invagination and protrusion of membrane to sequester cargoes within intraluminal vesicles; these cargoes are then degraded by lysosomal hydrolases.

**Figure 4 pathogens-13-00980-f004:**
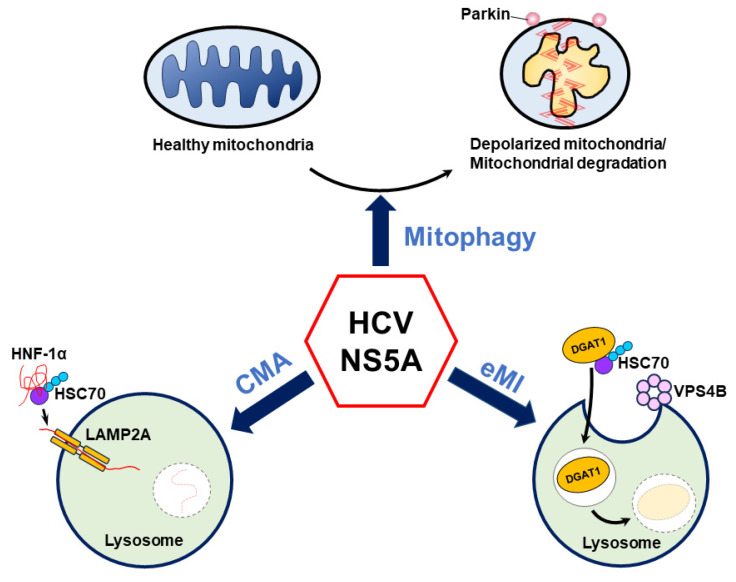
Current model of HCV NS5A-regulated autophagy. HCV NS5A induces mitochondrial depolarization and fission and triggers mitochondrial translocation to remove deformed mitochondria via mitophagy. Additionally, HCV NS5A acts as a bridge to connect HNF-1α and HSC70 on lysosomes, allowing the degradation of HNF-1α by LAMP2A-mediated CMA. In addition, HCV NS5A promotes the degradation of DGAT1 via endosomal microautophagy (eMI) through the binding of HSC70 and functional VPS4B.

**Table 2 pathogens-13-00980-t002:** Induction of autophagy by the HCV core, NS3, and NS4B proteins.

Approach	Phenotype	Functional Role	Reference
Transfection of NS4B in Huh7.5 cells	Activation of autophagy by NS4B transfection Increased formation of autophagic vacuoles induced by NS4B Activation of autophagy induced by NS4B requiring Rab5 and PI3K/Vps34 Binding of NS4B to Rab5, PI3K/Vps34, and Beclin 1	Promote viral RNA replication	[264]
Transfection of NS3 in Huh7.5 cells	Activation of autophagy by NS3 through immunity-associated GTPase family M (IRGM) Increased interaction between NS3 and IRGM Increased virus production in infected cells induced by IRGM	Promote viral RNA replication	[265]
Transfection of the core protein in Huh7 cells	Activation of autophagy induced by expression of the HCV core protein Increased autophagic flux induced by the HCV core protein Complete autophagy induced by the HCV core protein Requirement of EIF2AK3 and ATF6 in the UPR pathway for HCV core-activated autophagy	Unclear	[266]

**Table 3 pathogens-13-00980-t003:** Activation of autophagy by HCV NS5A.

Approach	Phenotype	Functional Role	Reference
Transfection of NS5A in Huh-7.5 cells	Formation of GFP-LC3-labeled autophagic vacuoles induced by HCV NS5A Complete autophagy induced by HCV NS5A Triggering of mitochondrial depolarization and fission by HCV NS5A Enhanced mitochondrial translocation of Parkin and activation of mitophagy induced by HCV NS5A Suppression of ROS levels to inhibit HCV-induced mitophagy	Counteract oxidative stress	[8]
Transfection of NS5A in Huh-7.5 cells/infection of Huh-7.5 cells with HCV J6/JFH1 (2a)	Increased interaction between HNF-1α and HSC70 Binding of HNF-1α to HSC70 through the putative pentapeptide ‶Gln-Arg-Glu-Val-Val″ Colocalization of HCV NS5A and HNF-1α in lysosomes HNF-1α degradation induced by HCV NS5A and HCV infection Suppression of HNF-1α degradation by LAMP2A gene knockdown in infected cells	Modulation of glucose metabolism	[9]
Transfection of NS5A in Huh-7.5 cells/infection of Huh-7.5 cells with HCV J6/JFH1 (2a)	Colocalization of HCV NS5A with DGAT1 Colocalization of HCV NS5A with DGAT1 The binding of DGAT1 to HSC70 through the pentapeptide “Gln-Val-Glu-Lys-Arg″ in DGAT1 DGAT1 degradation induced by HCV NS5A and HCV infection Suppression of DGAT1 degradation by VPS4B gene knockdown in infected cells	Regulation of lipid metabolism	[10]

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
