# Peer review of "Functional Role of Hepatitis C Virus NS5A in the Regulation of Autophagy"

_pathogens, 2024, doi:10.3390/pathogens13110980_

Round 1
Reviewer 1 Report
Comments and Suggestions for Authors
Title: Functional Role of hepatitis C virus NS5A in the regulation of autophagy
Type of manuscript: Review
General comments:
In this review the authors mention that there is evidence that shown that HCV NS5A induce autophagy to promote degradation of organelles and proteins. In this manuscript the authors do a comprehensive review of the role of the Hepatitis C Virus (HCV) NS5A protein in the regulation of autophagy. Specifically, they provide an overview of the current understanding on how HCV NS5A regulates autophagy to promote mitochondrial turnover and protein degradation. They also provide a summary of the regulatory mechanisms in these processes.
Overall, the paper is well organized and pleasant to read. The authors detailed the mechanistic underpinnings of NS5A-induced autophagy and its relevance to the HCV life cycle. However, there are general and specific comments that should be taken in consideration by the editor in order to accept this work in this journal.
- Doing a quick search in WoS platform there are 4 other reviews in this field using the keywords: Hepatitis C virus, NS5A and autophagy. All these reviews have been recentlyu published (2020 – 2021). Would be good to know how the authors differentiate this current review from those currently available.
- Some sections need more detailed explanation of experimental data to support the authors key claims, particularly regarding autophagy induction mechanisms. Also, the authors could attempt to connect NS5A´s autophagy and the pathogenesis of liver diseases such as hepatocellular carcinoma, for example. This may add more clinical relevance to this review.
- Regarding the conclusions and perspectives: Would be good that the authors provide specific proposals for future research. Perhaps new ideas regarding the development of novel antiviral therapies targeting NS5A o else.
- Specific inconsistencies or unclear sections in the text that need clarification (detailed below in the specific comments).
Specific comments:
- Line 46: I provide…the papers was written by two authors. Ç
- Line 19 – 21: In the abstract, the authors mention that there is increasing evidence that HCV NS5A may induce autophagy to promote degradation of organelles and proteins. I know it is important to keep the abstract short. However, the authors may provide a more detailed explanation about the specific role of NS5A that could be further explain in the introduction.
- Line 35: Chronic HCV infection is the leading cause of late-stage liver disease. A reference is needed.
- Line 44-45: “...implying that HCV NS5A could be a key player in the regulation of mitochondrial physiology and cellular metabolism in the infected cells…”. The authors need to add more specific examples or references detailing how NS5A’s interaction with mitochondria directly influences cell metabolism and authophagy.
- Line 52-62: HCV Epidemiology. The authors may provide more recent epidemiological data. They mention 2 papers that were published in 2003 and 2005 mentioning that over 3% of the population has been infected with HCV, generating a public health burden.
- Line 59: DAAs should be defined
- Line 118-125: The role of NS5A in virion assembly is presented concisely but would benefit from further explanation of how this process interacts with autophagy mechanisms, as mentioned later in the paper.
- Line 133-138: In the section discussing NS5A phosphorylation, the authors mention phosphorylation sites. To visualize this, would be useful to have a figure to show the relevant residues and their positions within NS5A.
-
- Line 238: Would be useful that the authors explain lipidation, also in several lines in the paper the word phagophore is mentioned and would be useful for the authors to explain this term.
-
Author Response
Dear Reviewer:
Thank you for giving us the opportunity to resubmit my manuscript entitled ‶Functional Role of Hepatitis C Virus NS5A in the Regulation of Autophagy″ to Pathogens, (Manuscript ID: pathogens-3276768). we appreciate the thoughtful and constructive suggestions provided by the reviewers. The content of this manuscript has been strengthened according to the reviewers’ comments. The revised manuscript shows the changes and point-by-point responses to each comment, which are listed below.
Point 1: Doing a quick search in WoS platform there are 4 other reviews in this field using the keywords: Hepatitis C virus, NS5A and autophagy. All these reviews have been recently published (2020 – 2021). Would be good to know how the authors differentiate this current review from those currently available.
Response 1: We thank the reviewer for the thoughtful comments on our manuscript. As the reviewer mentioned, there are some review articles showing HCV and general autophagy. However, these papers did not focus on the role(s) of HCV NS5A on the regulation of other forms of autophagy, including selective autophagy, which were demonstrated recently. In our manuscript, we aim to overview recent progress shown how HCV NS5A activates mitophagy, chaperone-mediated autophagy (CMA), and endosomal microautophagy (eMI). Moreover, we summarize the physiological significance of HCV NS5A’s regulation on these autophagic processes. On the other hand, most previous review articles on autophagy and HCV focus on briefly introducing the conclusions of different studies, and limited information on the detailed strategies and few comparisons between different papers are provided. We expect that this review article will provide comprehensive knowledge of the HCV NS5A-induced mitochondria turnover by mitophagy, the CMA-mediated degradation of hepatocyte nuclear factor 1 alpha (HNF-1a), and eMI-mediated diacylglycerol acyltransferase 1 (DGAT1) degradation (please see section 3.1~section 3.3; section 7.1~section 7.3; section 8). We intend to provide updated information on the molecular action of functional molecules involved and the implications of the clinical relevance of these studies. We believe our manuscript is quietly different from the previous studies and, importantly, allows readers to understand the functional role of HCV NS5A in regulating host cellular autophagy. Thank you again for the thoughtful suggestions.
Point 2: Some sections need more detailed explanation of experimental data to support the authors key claims, particularly regarding autophagy induction mechanisms. Also, the authors could attempt to connect NS5A´s autophagy and the pathogenesis of liver diseases such as hepatocellular carcinoma, for example. This may add more clinical relevance to this review.
Response 2: We are very grateful for the reviewer’s thoughtful comment. In the revised manuscript, we have incorporated the information on the autophagy induction by HCV and its regulatory mechanism is section 6.1. Please see lines 346-358 on paragraph 2 on page 9 in the revised manuscript. For the connection of HCV NS5A-regulated autophagy and HCV-related liver diseases, such as HCC, we have incorporated the discussion on the clinical relevance of HCV NS5A-regulated autophagy in liver diseases, including HCC and liver steatosis in section 7.1~section 7.3 and section 8 of the revised manuscript. Please see lines 429-439 on paragraph 1 on page 14, lines 467-471 on paragraph 1 on page 15, lines 494-499 on paragraph 2 on page 15, and line 554 on paragraph 2 on page 16 to line 569 on paragraph 1 on page 17 in the revised manuscript.
Point 3: Regarding the conclusions and perspectives: Would be good that the authors provide specific proposals for future research. Perhaps new ideas regarding the development of novel antiviral therapies targeting NS5A o else.
Response 3: We thank the reviewer for this comment. We have incorporated the information on the future research plan and the development of novel antiviral treatment in section 8 of the revised manuscript. Please see line 554 on paragraph 2 on page 16 to line 569 on paragraph 1 on page 17 in the revised manuscript.
Point 4: Specific inconsistencies or unclear sections in the text that need clarification. Line 46: I provide…the papers was written by two authors.
Response 4: We appreciate the reviewer for this correction. The “I” has been corrected to “we”. Please see line 22 on paragraph 1 on page 1 in the revised manuscript.
Point 5: Line 19 – 21: In the abstract, the authors mention that there is increasing evidence that HCV NS5A may induce autophagy to promote degradation of organelles and proteins. I know it is important to keep the abstract short. However, the authors may provide a more detailed explanation about the specific role of NS5A that could be further explain in the introduction.
Response 5: We appreciate the reviewer’s comment. We have revised the sections of “Abstract” and “Introduction” to describe the role(s) of HCV NS5A in regulating autophagy more detailed in the revised manuscript. Please see lines 20-24 on paragraph 1 on page 1, line 43 on paragraph 2 on page 1 to line 50 on paragraph 1 on page 2 in the revised manuscript.
Point 6: Line 35: Chronic HCV infection is the leading cause of late-stage liver disease. A reference is needed.
Response 6: We thank the reviewer for this comment. We have added the references in the " Introduction " section as the reviewer mentioned. Please see lines 36-37 on paragraph 1 on page 1 in the revised manuscript.
Point 7: Line 44-45: “...implying that HCV NS5A could be a key player in the regulation of mitochondrial physiology and cellular metabolism in the infected cells…”. The authors need to add more specific examples or references detailing how NS5A’s interaction with mitochondria directly influences cell metabolism and authophagy.
Response 7: We appreciate the reviewer for this suggestions. In section of “Introduction”, we have added the references shown the regulation of HCV NS5A on mitochondrial physiology and cellular metabolism. Please see line 43 on paragraph 2 on page 1 to line 50 on paragraph 1 on page 2 in the revised manuscript. For the detailed discussion on the information on how HCV NS5A’s regulation on mitochondrial physiological and cellular metabolism in sections 7.1~7.3 of the revised manuscript. Please see line 379 on paragraph 2 on page 12 to line 499 on paragraph 2 on page 15 in the revised manuscript.
Point 8: Line 52-62: HCV Epidemiology. The authors may provide more recent epidemiological data. They mention 2 papers that were published in 2003 and 2005 mentioning that over 3% of the population has been infected with HCV, generating a public health burden.
Response 8: Thank you very much for this suggestion. We have updated the references in section 2.1 “HCV epidemiology” as the reviewer mentioned. Please see lines 53-54 on paragraph 2 on page 2 in the revised manuscript.
Point 9: Line 59: DAAs should be defined
Response 9: We thank the reviewer for this suggestion. We have incorporated the definition of DAAs in the section “Introduction”. Please see lines 37-40 on paragraph 1 on page 1 and lines 59-64 on paragraph 2 on page 2 in the revised manuscript.
Point 10: Line 118-125: The role of NS5A in virion assembly is presented concisely but would benefit from further explanation of how this process interacts with autophagy mechanisms, as mentioned later in the paper.
Response 10: We appreciate the reviewer's suggestions. We have incorporated a further explanation of how the HCV NS5A-regulated autophagy potentially regulates the assembly of virion in section 8 of the revised manuscript. Please see lines 551-556 on paragraph 2 on page 16 in the revised manuscript.
Point 11: Line 133-138: In the section discussing NS5A phosphorylation, the authors mention phosphorylation sites. To visualize this, would be useful to have a figure to show the relevant residues and their positions within NS5A.
Response 11: We thank the reviewer for this comment. We have incorporated the information on phosphorylation sites of HCV NS5A in Figure 2 of the revised manuscript. Please see Figure 2 on page 4 and line 137 on paragraph 2 on page 4 in the revised manuscript.
Point 12: Line 238: Would be useful that the authors explain lipidation, also in several lines in the paper the word phagophore is mentioned and would be useful for the authors to explain this term.
Response 12: We are grateful to this suggestion. We have incorporated the definition of ATG8/LC3 lipidation and the characteristics of phagophore in sections of 4.2 and 4.3 in the revised manuscript. Please see line 208 on paragraph 1 on page 6, lines 221-223 on paragraph 1 on page 7, and lines 229-233 on paragraph 2 on page 7 in the revised manuscript.
We hope that this version of our manuscript and our responses address all your concerns and that this revised manuscript meets the criteria for publication in Pathogens. Thank you for your kind consideration.
Sincerely,
Po-Yuan Ke, Ph.D.
Associate Professor
Department of Biochemistry & Molecular Biology and Graduate Institute of Biomedical Sciences, College of Medicine, Chang Gung University, Taoyuan 33302, Taiwan, Republic of China
Liver Research Center, Chang Gung Memorial Hospital, Linkou, Taoyuan 33305, Taiwan, Republic of China
Tel: 886-3-2118800-5115
E-mail: pyke0324@mail.cgu.edu.tw

Reviewer 2 Report
Comments and Suggestions for Authors
It is an interesting manuscript about “Functional Role of Hepatitis C Virus NS5A in the Regulation of Autophagy”.
*HCV NS5A may induce autophagy to promote the degradation of organelles and proteins. HCV NS5A regulates autophagy to promote mitochondrial turnover and protein degradation.
The HCV nonstructural 5A (NS5A) protein was shown to induce host mitophagy, chaperone-mediated autophagy (CMA), and microautophagy, thus eliminating mitochondria, HNF-1and DGAT1, implying that HCV NS5A could be a key player in the regulation of mitochondrial physiology and cellular metabolism in infected cells.
HCV NS5A plays an unexpected role in regulating HCV-induced autophagy.
HCV NS5A induces autophagy to selectively eliminate mitochondria via mitophagy and target HNF-1and DGAT1 for CMA- and eMI-mediated degradation, respectively
HCV NS5A alone is sufficient to trigger autophagic degradation, which may allow cells to counteract oxidative stress, reduce glucose
metabolism, and alter LD biogenesis.
The HCV core protein has been shown to interrupt mitophagy in cells harboring HCV NS5A and to affect the LD localization of HCV NS proteins, whether HCV NS5A has a similar phenotype in infected cells
* Recently, autophagy has emerged as a cellular pathway, playing a role in several aspects of HCV infection. The molecular mechanisms that link the HCV life cycle with autophagy machinery is shown, especially the role of HCV/autophagy interaction in dysregulating inflammation and lipid homeostasis and its potential for translational applications in the treatment of HCV-infected patients.
*The interplay between autophagy and host innate immunity has been of great interest. Hepatitis C virus (HCV) impedes signaling pathways initiated by pattern-recognition receptors (PRRs) that recognize pathogens-associated molecular patterns (PAMPs). Autophagy, a cellular catabolic process, delivers damaged organelles and protein aggregates to lysosomes for degradation and recycling. Autophagy is also an innate immune response of cells to trap pathogens in membrane vesicles for removal. However, HCV controls the autophagic pathway and uses autophagic membranes to enhance its replication. Mitophagy, a selective autophagy targeting mitochondria, alters the dynamics and metabolism of mitochondria, which play important roles in host antiviral responses. HCV also alters mitochondrial dynamics and promotes mitophagy to prevent premature cell death and attenuate the interferon (IFN) response. In addition, the dysregulation of the inflammasomal response by HCV leads to IFN resistance and immune tolerance. These immune evasion properties of HCV allow HCV to successfully replicate and persist in its host cell.
HCV-induced autophagy/mitophagy and its associated immunological responses and how these processes are regulated in HCV-infected cells.
*Hepatitis C virus (HCV) is known to co-opt cellular autophagy pathway to promote its own replication. HCV regulates autophagy through multiple mechanisms to control intracellular protein and membrane trafficking to enhance its replication and suppress host innate immune response.
The interplay between HCV and autophagy and the crosstalk between HCV-induced autophagy and host innate immune responses is crucial..
*Above mentioned should be referred to.
Author Response
Dear Reviewer:
Thank you for giving us the opportunity to resubmit my manuscript entitled ‶Functional Role of Hepatitis C Virus NS5A in the Regulation of Autophagy″ to Pathogens, (Manuscript ID: pathogens-3276768). we appreciate the thoughtful and constructive suggestions provided by the reviewers. The content of this manuscript has been strengthened according to the reviewers’ comments. The revised manuscript shows the changes and point-by-point responses to each comment, which are listed below.
Point 1: Recently, autophagy has emerged as a cellular pathway, playing a role in several aspects of HCV infection. The molecular mechanisms that link the HCV life cycle with autophagy machinery is shown, especially the role of HCV/autophagy interaction in dysregulating inflammation and lipid homeostasis and its potential for translational applications in the treatment of HCV-infected patients.
Response 1: We appreciate the reviewer's thoughtful comments on our manuscript. We have strengthened the content of sections 6.1, 7.1~7.3, and 8 shown that the role(s) of HCV-induced autophagy on innate immunity, inflammation, and lipid metabolism. Please see line 316 on paragraph 4 on page 8 to line 345 on paragraph 1 on page 9, lines 429-439 on paragraph 1 on page 14, lines 467-471 on paragraph 1 on page 15, lines 494-499 on paragraph 2 on page 15, and line 554 on paragraph 2 on page 16 to line 569 on paragraph 1 on page 17 in the revised manuscript.
Point 2: The interplay between autophagy and host innate immunity has been of great interest. Hepatitis C virus (HCV) impedes signaling pathways initiated by pattern-recognition receptors (PRRs) that recognize pathogens-associated molecular patterns (PAMPs). Autophagy, a cellular catabolic process, delivers damaged organelles and protein aggregates to lysosomes for degradation and recycling. Autophagy is also an innate immune response of cells to trap pathogens in membrane vesicles for removal. However, HCV controls the autophagic pathway and uses autophagic membranes to enhance its replication. Mitophagy, a selective autophagy targeting mitochondria, alters the dynamics and metabolism of mitochondria, which play important roles in host antiviral responses. HCV also alters mitochondrial dynamics and promotes mitophagy to prevent premature cell death and attenuate the interferon (IFN) response. In addition, the dysregulation of the inflammasomal response by HCV leads to IFN resistance and immune tolerance. These immune evasion properties of HCV allow HCV to successfully replicate and persist in its host cell. HCV-induced autophagy/mitophagy and its associated immunological responses and how these processes are regulated in HCV-infected cells.
Response 2: We thank the reviewer for this suggestion. We have incorporated the information on the regulation of innate antiviral response and inflammasome by HCV-induced autophagy and mitophagy in the content of sections 6.1, 7.1~7.3, and 8 shown that the role(s) of HCV-induced autophagy on innate immunity, inflammation, and lipid metabolism. Please see line 316 on paragraph 4 on page 8 to line 345 on paragraph 1 on page 9, lines 429-439 on paragraph 1 on page 14, lines 467-471 on paragraph 1 on page 15, lines 494-499 on paragraph 2 on page 15, and line 554 on paragraph 2 on page 16 to line 569 on paragraph 1 on page 17 in the revised manuscript.
Point 3: Hepatitis C virus (HCV) is known to co-opt cellular autophagy pathway to promote its own replication. HCV regulates autophagy through multiple mechanisms to control intracellular protein and membrane trafficking to enhance its replication and suppress host innate immune response. The interplay between HCV and autophagy and the crosstalk between HCV-induced autophagy and host innate immune responses is crucial.
Response 3: We are grateful to this suggestion. We have strengthened the content of sections 6.1. shown the regulation of host immune response by HCV-activated autophagy (Please see line 316 on paragraph 4 on page 8 to line 345 on paragraph 1 on page 9 in the revised manuscript) and enhancement of viral replication and virion assembly by autophagy in HCV-infected cells (Please see line 346-362 on paragraph 2 on page 9) in the revised manuscript.
We hope that this version of our manuscript and our responses address all your concerns and that this revised manuscript meets the criteria for publication in Pathogens. Thank you for your kind consideration.
Sincerely,
Po-Yuan Ke, Ph.D.
Associate Professor
Department of Biochemistry & Molecular Biology and Graduate Institute of Biomedical Sciences, College of Medicine, Chang Gung University, Taoyuan 33302, Taiwan, Republic of China
Liver Research Center, Chang Gung Memorial Hospital, Linkou, Taoyuan 33305, Taiwan, Republic of China
Tel: 886-3-2118800-5115
E-mail: pyke0324@mail.cgu.edu.tw

Reviewer 3 Report
Comments and Suggestions for Authors
This review comprehensively summarizes the functional role of HCV NS5A in the regulation of autophagy. Here are some minor suggestions.
1. Relative references are suggested to be cited in the introduction. For example, line 32, 36 and 39.
2. The section of “4. Autophagy” is unnecessarily long.
Author Response
Dear Reviewer:
Thank you for giving us the opportunity to resubmit my manuscript entitled ‶Functional Role of Hepatitis C Virus NS5A in the Regulation of Autophagy″ to Pathogens, (Manuscript ID: pathogens-3276768). we appreciate the thoughtful and constructive suggestions provided by the reviewers. The content of this manuscript has been strengthened according to the reviewers’ comments. The revised manuscript shows the changes and point-by-point responses to each comment, which are listed below.
Point 1: Relative references are suggested to be cited in the introduction. For example, line 32, 36 and 39.
Response 1: We thank the reviewer for this comment. As the reviewer mentioned, we have added the references in the sentences of the “Introduction” section. Please see line 29 on paragraph 2 on page 1 to line 50 on paragraph 1 on page 2 in the revised manuscript.
Point 2: The section of “4. Autophagy” is unnecessarily long.
Response 2: We are grateful for this suggestion. In this review, we expect to provide comprehensive information on the interplay between HCV NS5A and autophagy, particularly for three forms of autophagy. In this manuscript, we intend to provide compelling information about the molecular regulation of all types of autophagy. As suggested by the reviewer, we have shortened and consolidated the content of sections 4.2, 4.4 5.1 and 5.2. Please see lines 222-224 on paragraph 1 on page 7, lines 253-258 on paragraph 3 on page 7, lines 279-286 on paragraph 2 on page 8, and lines 300-306 on paragraph 3 on page 7 in the revised manuscript. It is hoped that the reviewer kindly agrees with us to keep the review on autophagy regulation of the revised manuscript complete. Thank you again for the constructive comments.
We hope that this version of our manuscript and our responses address all your concerns and that this revised manuscript meets the criteria for publication in Pathogens. Thank you for your kind consideration.
Sincerely,
Po-Yuan Ke, Ph.D.
Associate Professor
Department of Biochemistry & Molecular Biology and Graduate Institute of Biomedical Sciences, College of Medicine, Chang Gung University, Taoyuan 33302, Taiwan, Republic of China
Liver Research Center, Chang Gung Memorial Hospital, Linkou, Taoyuan 33305, Taiwan, Republic of China
Tel: 886-3-2118800-5115
E-mail: pyke0324@mail.cgu.edu.tw
